# Research on the Influence of Coil LC Parallel Resonance on Detection Effect of Inductive Wear Debris Sensor

**DOI:** 10.3390/s22197493

**Published:** 2022-10-02

**Authors:** Heng Huang, Shizhong He, Xiaopeng Xie, Wei Feng, Huanyi Zhen

**Affiliations:** 1Guangzhou Mechanical Engineering Research Institute Co., Ltd., Guangzhou 510535, China; 2School of Mechanical & Automotive Engineering, South China University of Technology, Guangzhou 510640, China

**Keywords:** inductive wear debris sensor, debris, LC parallel resonance, inductance balance, induced electromotance

## Abstract

The coil structure of the inductive wear debris sensor plays a significant role in the effect of wear debris detection. According to the characteristics of LC parallel resonance, the capacitor and coil are connected in parallel to make sensor coils in the LC parallel resonance state, which is beneficial to improve the ability to detect wear particles. In this paper, the mathematical model of output-induced electromotance of the detection coil is established to analyze the influence of the structure on the detection sensitivity and enhance the sensor’s current rate of change to the disturbance magnetic field, which is essential to resist noise interference. Based on the coherent demodulation principle, the AD630 lock-in amplifier is applied to the test platform to amplify and identify weak signals. In addition, experiments are designed to test the output signals of debris under the condition of different original output voltages of the sensor with a parallel structure. Meanwhile, the near-resonance state of the detection coil with LC parallel circuit is tested by output signal information. Results show that the sensor detection sensitivity will be effectively improved with the LC parallel coil structure. For the sensor structure parameters designed in this paper, the optimal raw output amplification voltage for abrasive particle detection is 4.49 V. The detection performance of ferromagnetic particles and non-ferromagnetic particles is tested under this condition, realizing the detection ability of 103.33 μm ferromagnetic abrasive particles and 320.74 μm non-ferromagnetic abrasive particles.

## 1. Introduction

During the operation of mechanical equipment, friction pairs will wear out due to various factors such as stress and load, resulting in equipment failure and even serious accidents [1,2]. The debris in lubricating oil contains a lot of information about the operating status of mechanical equipment [3]. Therefore, abrasive particle detection is helpful to identify the operation status, realizing early warning and diagnosis in the early stage of equipment failure, in order to fundamentally reduce and avoid failures [4,5].

Wear debris sensors can be applied to monitor abrasive metal particles in the lubricant, while inductive wear debris sensors can be used to measure metal particle properties (ferromagnetic/non-ferromagnetic) and particle size range [6]. Three-coil inductive wear debris sensors are widely used to monitor abrasive particles in lubricating oil due to their detection characteristics. However, in actual use, the output-induced electromotance is relatively weak because of the weak signal generated by tiny abrasive particles, which is easily drowned among the external noise signal [7,8]. Therefore, further research is required to identify and monitor weak signals.

At present, relevant research on inductive wear debris sensors has been carried out all over the world. The successful applications are the TechAlert^TM^10 developed by MACOM Technologies in Lowell, MA, Usa, the MetalSCAN abrasive particle sensor developed by GasTOPS in Ottawa, Canada, and Type FG abrasive particle sensors developed by Kittwake in the Littlehampton, United Kingdom [9,10,11]. Among them, the most classic representatives are the Metal Scan series products developed by GasTOPS Company, which have been successfully applied in practical engineering. However, during actual use, because of various factors, weak signals generated by abrasive particles can be easily submerged by noise, resulting in unstable measurement and a high false positive rate [12]. Therefore, many studies have been carried out on how to improve the detection accuracy of sensors. Yonghui Yin studied the electromagnetic field of the single-coil inductive wear sensor and designed a sensor for experimental research [13]. Zhiwei Chen and Hongbo Fan et al. respectively researched the electromagnetic field of the double-coil structure, based on which they designed the inductive wear sensor. The monitoring tests on different types of abrasive metal particles were carried out, showing that the detection accuracy of iron and copper particles are 100 μm and 500 μm, respectively [14,15]. Huang Zhu and Shuhan Chen et al. surveyed the electromagnetic field of sensors with a three-coil structure. Zhu successfully applied the technology to the developed KLD-1 ferromagnetic particle quantifier, and Chen designed the sensor using the principle of electromagnetic induction, with the detection accuracy of iron particles being 100 μm [16,17].

Many studies have also been carried out on the monitoring and identification of weak signals generated by tiny abrasive particles in various industries. Conventional detection methods of weak signals are mainly based on the time and frequency domain [18]. However, certain limitations exist with all these methods, which are mainly manifested in the high signal-to-noise ratio threshold of weak signals that can be detected [19,20]. Donoho proposed a wavelet domain denoising technique based on the idea of threshold processing, which laid the foundation for sensors to reduce noise interference in 2015 [21,22,23]. Aiming at the problem that debris sensor wear is susceptible to background noise and vibration interference—which makes monitoring capability limited, resulting in false fault alarms—Bozchaloo filtered out vibration-related noise by applying a wavelet-based adaptation. Furthermore, wavelet threshold denoising was performed on the signal output from the adaptive filter to filter out the background noise mainly caused by cables and electronic components in the measurement system, which proved to be obvious in noise reduction by simulation and experiment [24].

It is found that the technical key of the wear debris sensor lies in the identification and detection of the signal generated by tiny abrasive particles through relevant research analysis. There are few studies on the influence of the original output-induced voltage of the three-coil inductive wear sensor on the detection accuracy at present. In order to improve the detection sensitivity of the sensor and increase the current change rate of the coil caused by abrasive particle disturbance, the LC parallel current resonance characteristic is applied to the sensor coil in this paper. The mathematical model of the sensor output-induced voltage under the LC parallel coil structure was established, and the characteristics of the output signal generated by abrasive particle disturbance under this structure were analyzed to improve the abrasive particle detection sensitivity. Through the test experiment, the optimum output-induced voltage value for the sensor abrasive particle monitoring was obtained by adjusting the LC parallel resonance state of the detection coil.

## 2. Principle Analysis of Inductive Wear Particle Sensor

The three-coil inductive wear debris sensor is mainly composed of two excitation coils and one detection coil, as shown in Figure 1. Excitation coils on both sides are used to provide the signal source, while the detection coil in the middle is for signal output. The detection principle is based on the magnetization effect and eddy current effect produced by particles in the magnetic field. Relevant properties of metal particles are identified by detecting the magnitude and phase of the output-induced electromotance.

Sinusoidal alternating currents $I_{1}$ and $I_{2}$ are applied to the two excitation coils. Based on the electromagnetic induction principle, the magnetic fields excited by two excitation coils are equal in magnitude and opposite in direction. Magnetic fluxes passing through the detection coil are equal because of the magnetic fields excited by two excitation coils when no debris flows through the sensor; as a result, the detection coil is in the state of inductance balance. According to Faraday’s law of electromagnetic induction, the output-induced electromotive force of the detection coil can be expressed as: $E=\frac{dQ}{dt}=0$. As there are abrasive particles passing through the sensor, the magnetic field generated by the abrasive particle disturbance to the coil is coupled with the original magnetic field due to the combined influence of the magnetization effect and the eddy current effect, leading to magnetic flux changes inside the detection coil. The magnetic flux passing through the detection coil is not 0, resulting in the corresponding induced electromotance. According to the relevant information of the induced electromotive force output by the detection coil, abrasive particle characteristics can be identified and judged.

## 3. Mathematical Model of Coil with LC Parallel Structure

LC parallel circuit is used to connect the inductor and capacitor in parallel in the circuit. For an inductive wear debris sensor, each coil can be equivalent to an inductance. The LC parallel state is constituted as the capacitance, and the coil is connected in parallel, which will be in the resonant state by adjusting the value of the inductor and capacitor. Due to the high-resistance characteristics of the resonant circuit, impedance changes caused by abrasive particles will be amplified. At the same time, the resonant circuit with frequency selection function can be applied to improve the anti-interference ability to other frequency noises. Therefore, based on the characteristic that the current change rate near the operating frequency can be increased due to the LC parallel resonant circuit, the excitation coil and the detection coil are connected in parallel with the capacitor to form the LC parallel circuit, which is beneficial to generate a larger magnetic flux difference between the two excitation coils when abrasive particles pass through, thereby increasing the mutual inductance voltage between the excitation coil and the detection coil and enhancing the output signal. The inductive wear debris sensor is designed based on the LC parallel resonance principle, and the equivalent circuit diagram is shown in Figure 2.

An AC current source $I=50^{-3}\cos(\omega t)A$ is applied to the excitation coil. Since each coil is in a synchronous resonance state, the sensor coil inductance $L_{qi}$, the resonant capacitance $C_{i}$, and the excitation signal frequency $f_{i}$ satisfy the resonance condition that:(1)$$f_{i}=\frac{1}{2\pi \sqrt{L_{qi}C_{i}}}$$

When abrasive particles pass through the sensor, the impedance of the excitation coil will change, resulting in a difference between the impedances of the two excitation coils, and the impedance of sensor excitation coil 1 can be written as:(2)$$Z_{1}=\frac{\left(j\omega L_{q1}+r\right)\frac{1}{j\omega C_{1}}}{\left(j\omega L_{q1}+r\right)+\frac{1}{j\omega C_{1}}}$$

As described in Equation (2), the sensor coil inductance can be shown as: $L_{q1}=L_{1}-M_{12}$. $L_{1}$ is the self-inductance of excitation coil 1, and $M_{12}$ represents the mutual inductance between the two excitation coils. As the two excitation coils are in the parallel resonance state, the circuit impedance value is maximum and much larger than that of the excitation coils. Coil inductance will change as abrasive particles transit through the coil, and the change value is written as ΔL. If the passing particles are ferromagnetic, the change value satisfies ΔL>0; while if the passing particles are non-ferromagnetic, the change value meets the equation ΔL<0. At this time, the inductance of excitation coil 1 can be expressed as: $\Delta L_{1}=L_{q1}+\Delta L$. As a result, the impedance difference between the two excitation coils as particles pass through the coil can be shown as:(3)$$\Delta Z=\frac{j\omega \Delta L}{\left(1-\omega^{2}C_{i}(L_{i}+\Delta L)+jC_{i}r\omega )(1-\omega^{2}C_{i}L_{i}+jC_{i}r\omega \right)}$$

While for the non-resonant state, the impedance $\Delta Z^{\prime }$ between excitation coils 1 and 2 caused by passing particles is the same as the impedance change $\Delta Z_{L}$ of the coil, namely, $\Delta Z^{\prime }=\Delta Z_{L}=j\omega \Delta L$. Besides, because of the resonance state, the expression satisfied $\omega^{2}CL\approx 1$. In conclusion, the impedance changes of the excitation coil in the resonant state and that in the non-resonant state caused by particles can be shown as:(4)$$\Delta Z=\frac{1}{-C_{i}{}^{2}r^{2}\omega^{2}-j\omega^{3}C_{i}{}^{2}\Delta Lr}\Delta Z^{\prime }$$

Taking ferromagnetic particle detection as an example, impedance changes of the excitation circuit caused by ferromagnetic particles with size ranging from 0 to 400 microns is shown in Figure 3a. The enlarged view of $\Delta Z^{\prime }$ is shown in Figure 3b for more details.

It can be seen from the figure that when debris goes through the sensor coil, the impedance change of the excitation coil in the LC parallel resonance state is much more obvious than that in the non-resonant state. Impedance changes are more pronounced as particle size increases. It shows that the impedance change of the coil will be more obvious, and the magnetic flux changes of the coil will be increased under the condition of the LC parallel resonance state, leading to the higher output-induced electromotance and more visible output signal, which is conducive to signal extraction and identification.

## 4. Test and Analysis

### 4.1. Excitation Coil LC Parallel Resonance Parameters

The resonant frequency of the sensor excitation coil is firstly tested by the experimental method in this part. As shown in Figure 4, a Keysight E4990A Impedance Analyzer (Santa Rosa, CA, USA) was applied to measure the relevant parameters of the excitation coils and detection coil, which can provide a high-precision measurement of impedance parameters for coils, including parameters such as conductance, inductance, and capacitance.

Under the condition that the input frequency is 150 kHz, sensor coils are in the non-resonant state, coil parameters of the inductive wear debris sensor are tested by the Impedance Analyzer, and the results are shown in Table 1.

We connected the synonyms of the two excitation coils to determine the parallel inductance value, as shown in Figure 5. Under the influence of the mutual inductance of the two coils, we found that the parallel inductance of the excitation coil was 55.514 μH under the condition of 150 kHz.

According to the measured inductance value, based on the relevant theories of parallel resonant, we calculated that the 20.162 nF capacitor and the excitation coil were applied to form the LC parallel resonant circuit. The frequency of the excitation coil LC parallel resonance was measured by the Impedance Analyzer. On the basis of characteristics that the impedance value is maximum with the LC parallel circuit resonates, the frequency of the excitation coil LC parallel resonance was measured to be 150.26 kHz, as shown in Figure 6.

### 4.2. Abrasive Particle Signal Extraction

A sinusoidal AC signal was applied to the excitation coils. Magnetic flux of the excitation coils changes due to the electromagnetic effect as abrasive particles pass through the sensor. Because of the mutual inductance between the coils, the relevant signals generated by abrasive particles are transmitted to the detection coil and output in the form of induced electromotance, which is similar to the signal demodulation process [25]. Using the multiplier, a reference signal coherent with the carrier frequency (same frequency and phase) was input and multiplied with the carrier frequency. After low-pass filtering, the low-frequency component of the signal was output. On the basis of this method, relevant signals of abrasive particles can be extracted from the output-induced electromotive force.

The equivalent inductance $L_{q3}$ of the detection coil will be changed as debris flows through the coil. According to the phase difference in the LC parallel circuit, the phase of the induced electromotive force output by the detection coil was also changed and can be described as:(5)$$\theta =\arctan\frac{2\pi fL_{q3}-\frac{1}{2\pi fC_{3}}}{R_{3}}$$

Based on the principle of coherent demodulation, assume that the detection signal carrying the abrasive particle information and the reference signal are, respectively, $U_{0}$ and $U_{1}$. They are shown as follows:(6)$$U_{0}=U_{m0}\sin\left(2\pi ft+\theta \right)$$
(7)$$U_{1}=U_{m1}\sin\left(2\pi ft\right)$$

Correlating the two signals through the multiplier to obtain the signal $U_{2}$ can be expressed as:(8)$$U_{2}=U_{0}U_{1}=\frac{1}{2}U_{m0}U_{m1}\cos\theta -\frac{1}{2}U_{m0}U_{m1}\cos\left(4\pi ft+\theta \right)$$

Passing the processed signal through the low-pass filter to filter out the high-frequency components in the signal, the low-frequency component of the signal obtained can be written as:(9)$$U_{out}=\frac{1}{2}U_{m0}U_{m1}\cos\theta$$

If the detection signal $U_{0}$ carrying abrasive particle information is in the same phase as the reference signal $U_{1}$, that is θ=0, the output signal value is maximum with the signal processed by the multiplier and low-pass filter. In order to obtain the signal information of abrasive particles as much as possible, on the basis of the principle of coherent demodulation, the reference signal performs a certain angle compensation by means of phase shift to reduce the phase difference between the voltage generated by the abrasive particle signal and the reference voltage, leading to obtaining the maximum DC output voltage.

### 4.3. Test Platform Design

The test platform is designed and constructed to analyze the detection performance of the three-coil wear debris sensor according to the above analysis of the LC parallel resonance structure. A schematic diagram of the test platform structure is shown in Figure 7. The test platform consists of a signal generator, an LC parallel loop composed of sensor coils and capacitors, a lock-in amplifier, and an oscilloscope. The coil and capacitor are connected in parallel to form an LC parallel resonance state to improve the detection sensitivity. The detection of the weak signal generated by tiny abrasive particles is carried out by the lock-in amplifier. Signals caused by abrasive particles are observed through the oscilloscope. The physical map of the test platform is shown in Figure 8.

### 4.4. Test and Analysis

On the basis of the test platform, according to the analysis of the influence of the LC parallel resonant circuit on the detection accuracy, the sensor detection results on abrasive particles under different conditions are analyzed. In view of the analysis and calculation, the value of the capacitance connected in parallel with the detection coil was 7.534 nF. For the purpose of increasing output-induced electromotance changes and facilitating the observation of the output signal, the output voltage was amplified by 50 times to test the detection effect of abrasive particles under different conditions.

#### 4.4.1. Influence of Parallel Capacitance on Particle Detection

Firstly, the particle detection ability of the coil with shunt capacitance or not was analyzed. Particles applied in the experiment were standard spherical. The material of ferromagnetic particles was iron, and the material of non-ferromagnetic particles was copper. Particle size was calibrated by the electron microscope at the beginning of the test. According to the calibrated size range, ferromagnetic and non-ferromagnetic particles in each size range were selected for detection. Each set of tests was conducted using an individual particle. Under the condition of inductance balance, the detection coil and excitation coils were connected in parallel with capacitance to put them in a parallel resonance state. Compared with the coil without shunt capacitors, the ferromagnetic particles with a size of 195.24 μm passing through the sensor were designed to detect the difference in the induced electromotance.

A relevant signal could be sensed by the detection coil due to the mutual inductance between the coils as abrasive particles pass through the sensor, which is output in the form of induced electromotance. Test results of abrasive particles by the sensor of coils with parallel capacitance or not are shown in Figure 9a,b. Comparing the two figures, we found that the peak-to-peak value of the sensor’s output-induced electromotance was about 0.01 mV if there was no parallel capacitance with connected coils as ferromagnetic particles transit through the sensor. Meanwhile, when coils are connected to the capacitance and in the resonance-sensitive state, the peak-to-peak value of the sensor’s output-induced electromotance was about 0.025 mV, significantly higher than that with no parallel capacitor, which shows that the parallel capacitance can greatly enhance the output signal caused by abrasive particles and improve the detection performance of the sensor.

#### 4.4.2. Influence of Different Original Output Voltages of Detection Coil in LC Parallel Resonance State

We adjusted the coil parameters and parallel capacitance to put both the excitation coils and detection coil in the near-resonance state. Different original output voltages of the detection coils can be obtained by adjusting the relative positions of the detection coil and excitation coils. Ferromagnetic particles were applied in the test, and the particle size was 195.24 μm calibrated under the electron microscope. Different original output voltages were set in the test conditions. Calibrated particles were carried through the sensor, with output signals shown on the oscilloscope screen. The test results are shown in Figure 10.

As can be seen from the above figures, when ferromagnetic particles go through the sensor, the magnetic flux of the detection coil changes due to the combined influence of the electromagnetic effect and the eddy current effect, leading to output-induced electromotance changes in the detection coil. Under the condition that the coil is in an LC parallel resonance state, the detection sensitivity to abrasive particles varies at different original output voltages of the detection coil, causing the output-induced electromotance change. The peak-to-peak value of the output-induced electromotance in the test was counted, and the obtained results are shown in Table 2.

It can be seen from the table that with the increase of the original output voltage of the detection coil, the output-induced electromotance shows a trend of first increasing and then decreasing as abrasive particles go through the sensor. As the original output amplified voltage of the sensor was 4.49 V, the measured peak-to-peak value of the detected DC output voltage was the maximum (74.35 mV). When the original output amplified voltage was more or less than 4.49 V, the peak-to-peak value of the output-induced electromotive force was reduced. Therefore, for the sensor coil designed in Section 4.1, the most suitable original output voltage was 4.49 V.

#### 4.4.3. Analysis of the Detection of Abrasive Particles under the Condition of Optimal Original Output Voltage

According to the test results in Section 4.3, the original output voltage of the detection coil was set to 4.49 V to test the detection effect on ferromagnetic and non-ferromagnetic particles. Particles of different sizes were calibrated under the electron microscope, passing through the sensor at the speed of 0.1 m/s. The output signal of the sensor was detected. Output-induced electromotance changes are shown respectively in Figure 11 and Figure 12 as ferromagnetic and non-ferromagnetic particles with different sizes pass through the sensor.

It can be seen from the figure that the internal magnetic field of the sensor will be changed as debris flows through the sensor under the condition that coils are in the LC parallel resonance state, resulting in induced electromotance changes. Under the premise that the original output voltage of the detection coil was 4.49 V, induced voltage changes are the most obvious due to the combined influence of electromagnetic effects and eddy current effects as abrasive metal particles pass through the sensor. Furthermore, larger particles refer to stronger signals. Meanwhile, because of the interference of the environment and sensor internal factors, noise signals will be generated along with output signals. Environmental factors include interference such as vibration, noise, and electromagnetic fields, while internal factors are mainly caused by the influence of electronic components. In this case, weak signals produced by tiny particles may be annihilated in the noise signal of the system, resulting in detection errors.

For metal particles, the magnetization effect and eddy current effect caused by particles of different sizes in the magnetic field are different, which means that output signals are miscellaneous as particles pass through the sensor. In regard to the LC parallel resonant structure designed in this paper, under the condition that the original output voltage is 4.49 V, obvious output-induced electromotance will be generated for particles larger than 100 μm. Moreover, the output signal strengthened with particle size increasing. As a consequence, the properties of ferromagnetic particles can be detected based on the amplitude of the output signal. While for non-ferromagnetic particles, the eddy current effect caused by the magnetic field is dominant and relatively small. Therefore, the output-induced electromotance will evidently be changed with particles larger than 300 μm. Similarly, for non-ferromagnetic particles with larger particle sizes, the output-induced electromotance is more obvious. Comparing Figure 11 and Figure 12, it can be found that under the condition of the original output voltage of 4.49 V, the detection effect of the sensor on ferromagnetic and non-ferromagnetic particles is pretty good.

## 5. Conclusions

Aiming at the three-coil inductive wear debris sensor, the detection performance for metal particles with the LC parallel resonance structure is studied in this paper. The test platform is built to test the most suitable original output voltage. The detection effect for ferromagnetic and non-ferromagnetic particles under this condition is tested and verified. The conclusions are as follows:(1)The impedance changes of coils will be more obvious as coils are in the LC parallel resonance state, leading to the increasing magnetic flux changes. The output-induced electromotance is increased, which is beneficial to the extraction and identification of the output signal.(2)Assuming that the sensor coil is connected in parallel with capacitors, compared with the case without capacitors in parallel, the output signal caused by debris passing through coils will be greatly enhanced, causing the detection performance evidently improved.(3)The original output voltage of the sensor has a great impact on the debris detection sensitivity. Signal changes caused by particles passing through the coil are most pronounced at the original output amplification voltage of 4.49 V. Meanwhile, when the output voltage is more or less than 4.49 V, the signal change gradually weakens.(4)Under the condition that the original output voltage is 4.49 V, the sensor is in the LC parallel resonance state, showing a good detection effect on metal particles. For ferromagnetic particles with a size of 100 μm or more and non-ferromagnetic particles with a size of 300 μm or more, the output signal of the sensor is obvious and easily identifiable.

## Figures and Tables

**Figure 1 sensors-22-07493-f001:**
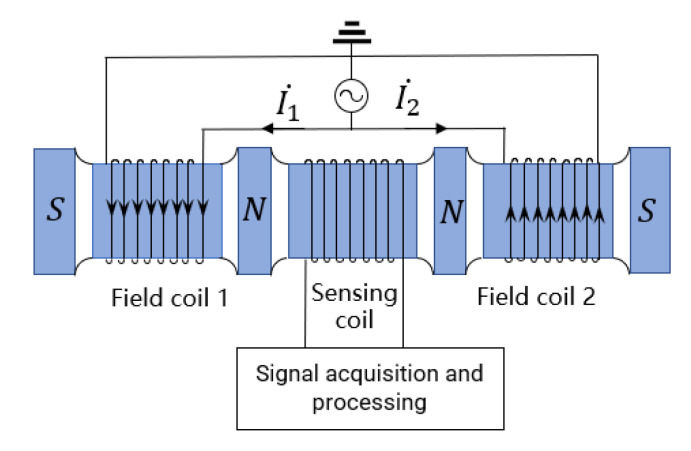
Schematic of detection.

**Figure 2 sensors-22-07493-f002:**
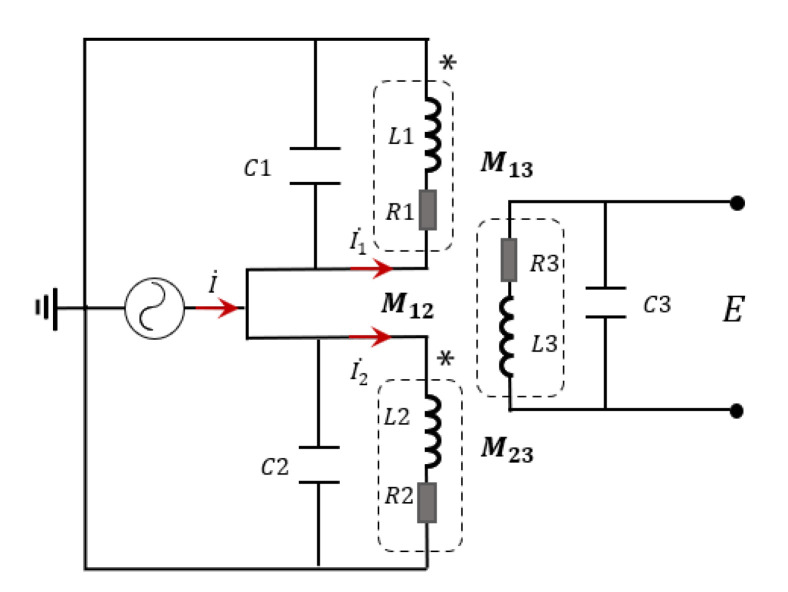
Equivalent circuit diagram of three coils.

**Figure 3 sensors-22-07493-f003:**
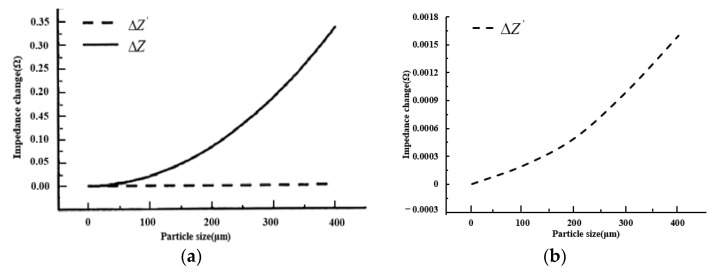
Test results. (**a**) Result comparison; (**b**) enlarged view.

**Figure 4 sensors-22-07493-f004:**
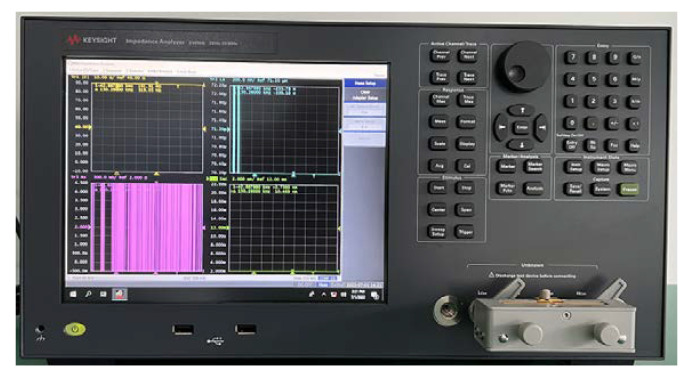
Keysight E4990A Impedance Analyzer.

**Figure 5 sensors-22-07493-f005:**
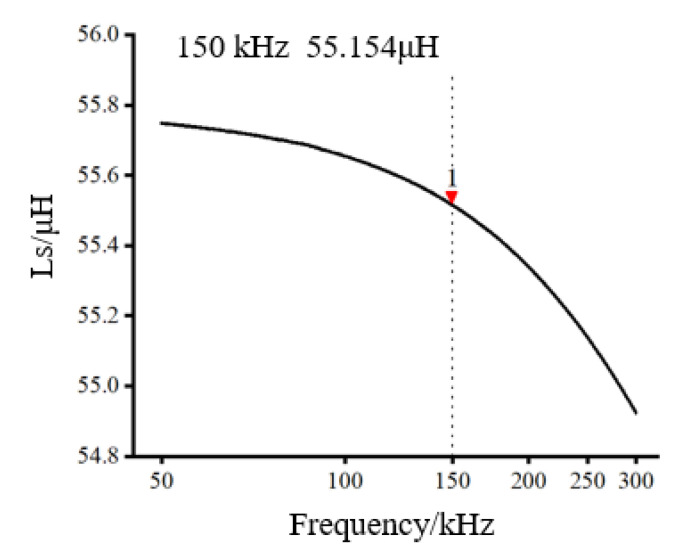
Excitation coil inductance.

**Figure 6 sensors-22-07493-f006:**
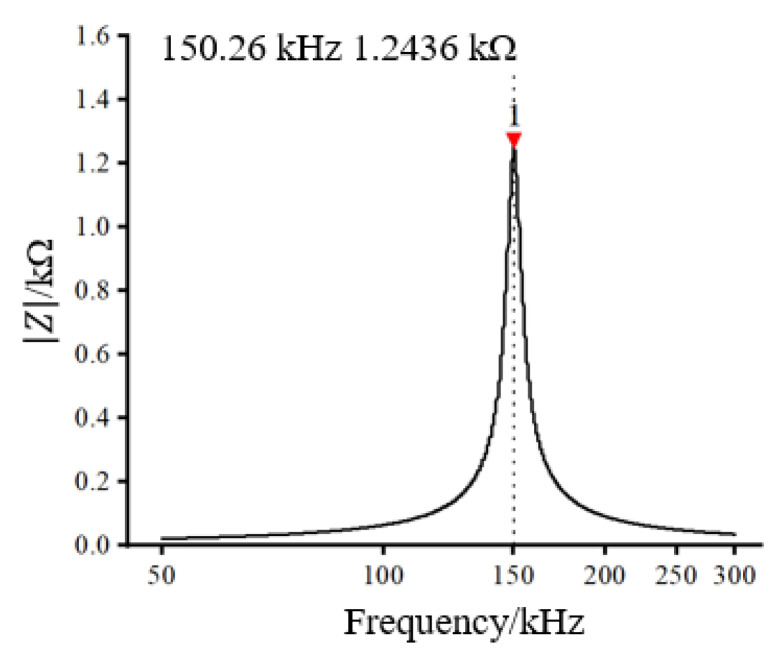
Resonant frequency measurement results.

**Figure 7 sensors-22-07493-f007:**
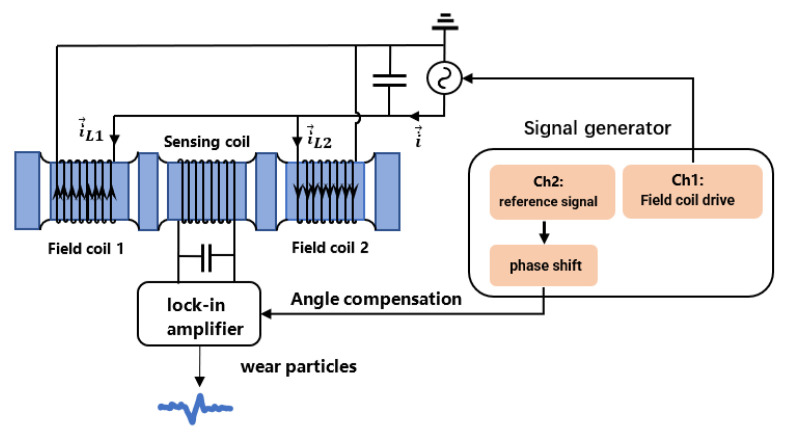
Schematic diagram of the test platform.

**Figure 8 sensors-22-07493-f008:**
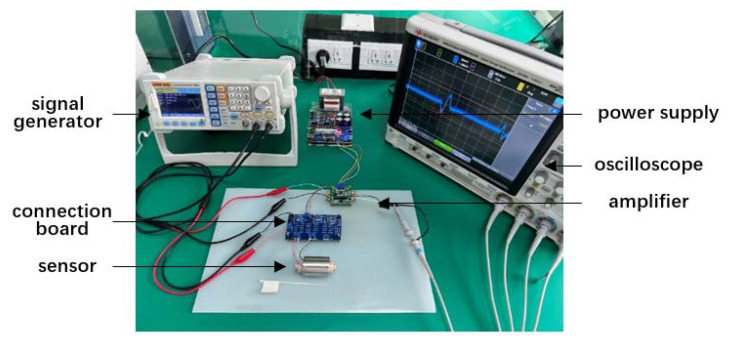
Physical map of the test platform.

**Figure 9 sensors-22-07493-f009:**
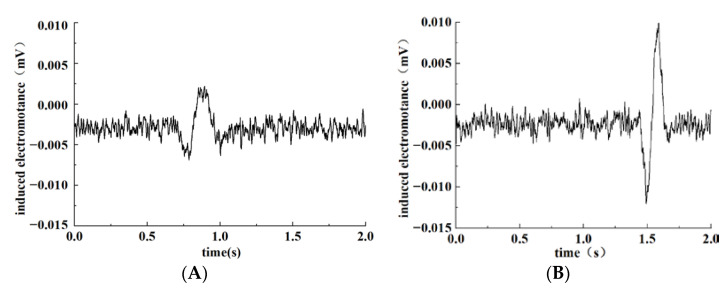
Influence of LC parallel capacitance on particle detection. (**A**) No capacitor in parallel; (**B**) capacitor in parallel.

**Figure 10 sensors-22-07493-f010:**
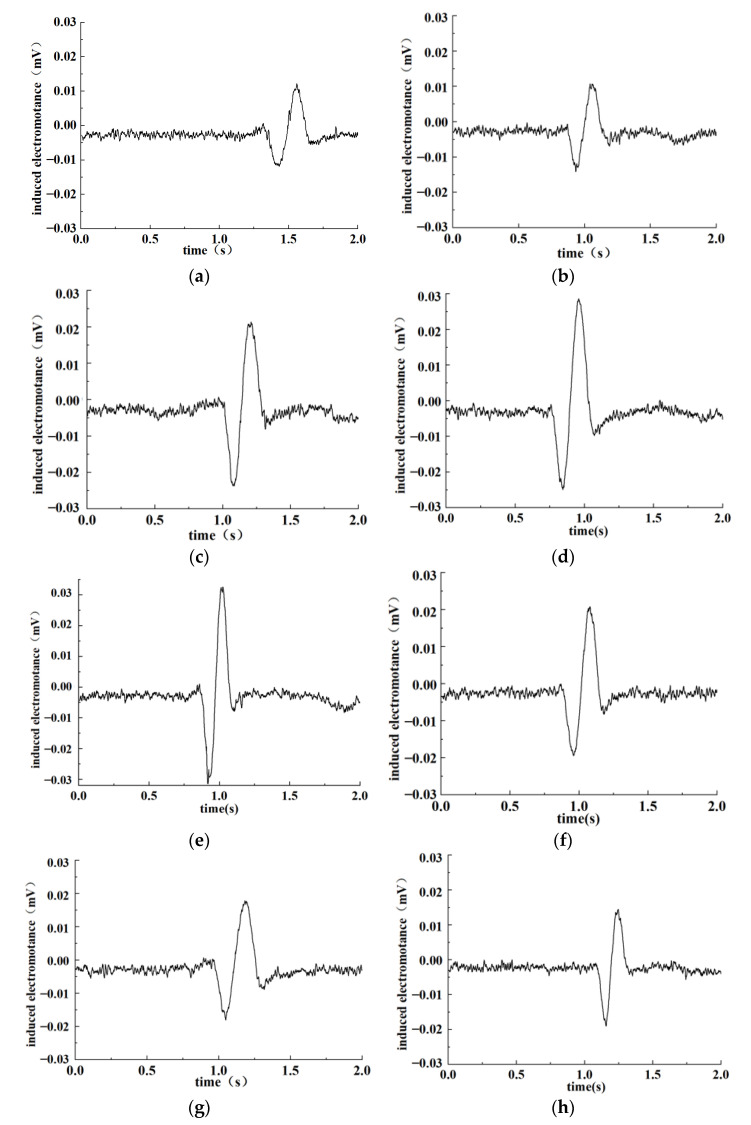
Test results under different raw output voltage conditions. (**a**) 3.1 V; (**b**) 3.62 V; (**c**) 3.9 V; (**d**) 4.3 V; (**e**) 4.49 V; (**f**) 4.63 V; (**g**) 4.8 V; (**h**) 5 V.

**Figure 11 sensors-22-07493-f011:**
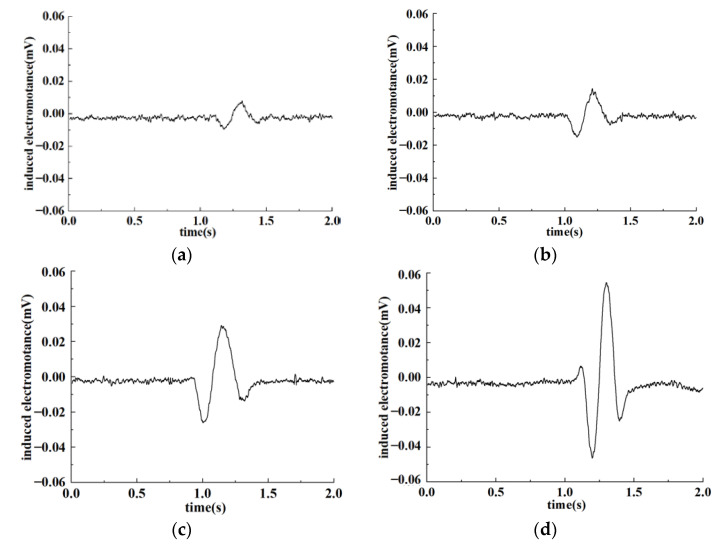
Detection results of ferromagnetic particles. (**a**) 103.33 μm; (**b**) 134.59 μm; (**c**) 167.1 μm; (**d**) 250.53 μm.

**Figure 12 sensors-22-07493-f012:**
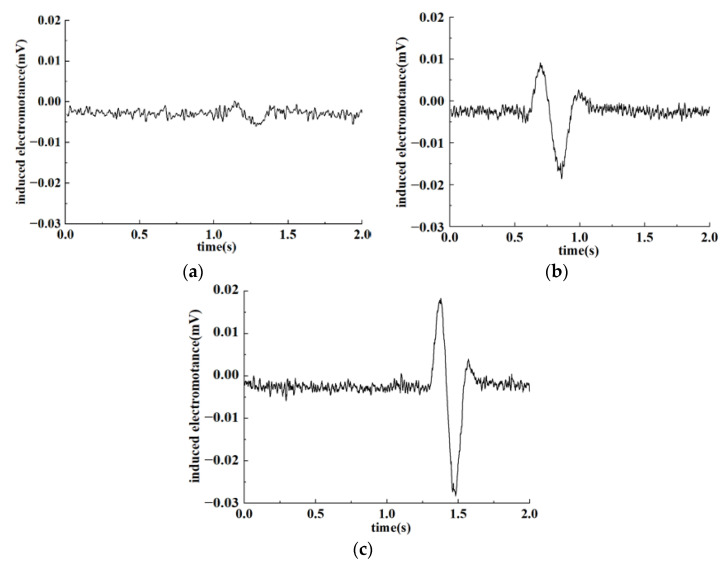
Detection results of non-ferromagnetic particles. (**a**)3 20.74 μm; (**b**) 458.72 μm; (**c**) 550.04 μm.

**Table 1 sensors-22-07493-t001:** Coil parameters.

Coil	Inner Diameter/mm	Turns Number	Length/mm	Inductance/uH	Resistance/Ω	Excitation Voltage/V
Excitation coil 1	9	108	5	114.75	4.303	20
Detection coil 1	9	135	7	152.07	5.455	/
Excitation coil 1	9	108	5	114.68	4.325	20

**Table 2 sensors-22-07493-t002:** Detection results under different original output amplification voltages of detection coils.

Output Amplification Voltage/V	Output Induced Electromotance (p-p)/mV
3.100	35.675
3.620	36.000
3.900	57.675
4.300	66.575
4.490	74.350
4.630	52.000
4.800	47.000
5.000	43.750

## Data Availability

The data that support the findings of this study are available from the corresponding author upon reasonable request.

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
