# Peer review of "Research on the Influence of Coil LC Parallel Resonance on Detection Effect of Inductive Wear Debris Sensor"

_sensors, 2022, doi:10.3390/s22197493_

Round 1

Reviewer 1 Report

The topic is intriguing. However, the presentation of the paper is not good. Some typos can be found anywhere, and the authors do not adhere to the journal format. My specific remarks are listed below.

1.      The figures are too small if compared to the margin. Also, please replace the figures with higher resolution

2.      Figure 11-15. Include a discussion of why there is noise. What was the cause?

3.      Table 2. Use consistent writing for a significant number. In certain instances, authors use one number after the point, whereas in other instances, they use three numbers after the bullet. Be consistent to avoid misunderstanding.

Round 2
